# Femtosecond Laser Line-by-Line Inscribed Seven Core Fiber Cascaded Fabry–Perot Cavity and Its Vectorial Bending Sensing Application

**Yanqing Zhang** [1,†], **Haili Ma** [2,†], **Yicun Yao** [1,*], **Minghong Wang** [1,*], **Liqiang Zhang** [1], **Zhaogang Nie** [1] **and Chenglin Bai** [1]

1    Key Laboratory of Optical Communication Science and Technology of Shandong Province, School of Physical Science and Information Technology, Liaocheng University, Liaocheng 252059, China
2    School of Medicine, Liaocheng University, Liaocheng 252000, China
\*    Correspondence: yaoyicun@lcu.edu.cn (Y.Y.); wangminghong@lcu.edu.cn (M.W.)
†    These authors contributed equally to this work.

**Abstract:** Multi-core fibers have been widely used for vector-bending sensing due to their off-axis distributed cores. In contrast to vector-bending sensors based on Bragg gratings, fiber Fabry–Perot (F–P) interferometers are more advantageous due to their ease of fabrication and potential for introducing the Vernier effect to further improve sensitivity. We propose and experimentally demonstrate a cascaded Fabry–Perot (F–P) cavity vector bending sensor. From the experimental results, the sensor has a strong bending dependence with a maximum sensitivity of 123.12 pm/m$^{-1}$, and the curvature magnitude and direction can be reconstructed from the tilted wavelength shift of the asymmetric fiber-core F–P cavities.

**Keywords:** femtosecond laser; Fabry–Perot (F–P); vector bending; SCF

## 1. Introduction

Optical fiber sensors are extensively employed in diverse fields due to their unique advantages, such as compact structure, high sensitivity, and resistance to electromagnetic interference [1–5]. In recent years, fiber optic bending sensors have been widely applied in deformation monitoring, mechanical engineering, robotics, and other fields [6–8]. One-dimensional bending sensors, which can only identify positive and negative directions, can be achieved by breaking the cylindrical symmetry of the fiber through introducing off-axis fiber Bragg gratings (FBGs), tilted FBGs, asymmetric long-period fiber gratings (LPGs), and lateral-offset Mach–Zehnder Interferometers (MZIs), for instance [9–13]. Compared to one-dimensional bending sensors, two-dimensional fiber bending sensors, also known as fiber vector bending sensors, can determine both the curvature radius and the bending directions simultaneously [14–16], thus having greater significance in practical applications.

On the other hand, multi-core fibers (MCFs) have increasingly attracted the attention of researchers in the field of optical communications, due to exponential growth in data transmission requirements. Thanks to the special cross-sectional geometry of MCFs, multiple, one-dimensional bending sensors can be integrated into the single MCF [17–19], thus constructing a compact and robust fiber vector bending sensor. In 2022, Qi et al. proposed construction of the vectorial bending sensor by splicing a section of quartz capillary fiber between a seven-core fiber (SCF) and a multimode fiber, but with complex structure and high insertion loss [20]. In 2018, Hou et al. proposed inscribing FBG in each core of a SCF and investigated the vector bending response of fiber gratings with six outer cores in a 360° direction [21]. In 2020, Zhu et al. proposed a stress-insensitive vector curvature sensor based on a single FBG. However, the fiber Bragg grating needs to be packaged on a thin steel plate and coated with UV glue, and this preparation is cumbersome [22].

Compared with FBG, cascaded Fabry–Perot (F–P) resonators are simple to prepare and require low processing accuracy [23]. In 2022, Olieira et al. proposed the use of resin-based F–P interferometers to manufacture and characterize temperature-insensitive two-dimensional curvature sensors with sensitivities greater than 400 pm/m$^{-1}$. The precise control of the photopolymerizable resin cavity length and diameter of the resin section may be difficult, which limits the reproducibility of the sensor device [24]. Very recently, Yang et al. reported that a femtosecond laser plane-by-plane inscribed seven-core fiber FPs, in which the vernier effect was introduced with a record high bending sensitivity of 4.900 nm/∘ realized for fiber F–P interferometers. However, the post-data fitting procedure is needed for a Vernier-based fiber sensor, which will cause longer demodulation and is time consuming [25]. Additionally, fitting errors may exist, considering the low resolution of the Vernier envelope.

In this study, we used femtosecond laser line-by-line inscription technique to inscribe multiple sets of reflectors in different cores of a seven-core fiber, to form a cascaded F–P structure. Since the reflectors within each core respond differently to vector bending, the bending direction and curvature magnitude can be reconstructed from the spectral displacement of any two non-diagonal outer core fiber gratings. The line-by-line process can guarantee fast processing with the total processing time for the fabrication of each cascaded FPs to be controlled below 10 s, which shows potential importance in the application point of view. The sensor is simple in structure, easy to fabricate, and has strong reproducibility, thus, it has a wide range of application prospects.

## 2. Principle and Preparation of the Sensor

The working principle of a fiber cascaded F–P sensor can be understood as shown in Figure 1a, in which a number of equally spaced internal mirrors along the fiber core are arranged perpendicular to the fiber axis. The incident beam traveling along the fiber core is reflected by each internal reflector in turn, and the resulting reflected beams are combined to form the interference. The electric field reflectivity of each reflector can be expressed as:

$$r = \left| \frac{n_1 - n_2}{n_1 + n_2} \right| \tag{1}$$

in which $n_1$ is the refractive index of the reflector, and $n_2$ is the effective refractive index of the fundamental core mode.

The propagation phase delay of two neighboring reflectors can be expressed as:

$$\phi = 2n_2 k_0 L \tag{2}$$

where $L$ is the cavity length of F–P.

Assuming that the electric field of the incident beam is $E_0 = 1$, and we take the transmittance of each mirror as 100% approximately considering the low value of $r$, the number of mirrors is set to be $N$. Then, we obtain the reflected electric field from the mirrors $M_1$, $M_2$, $M_3$ ... and $M_n$ to be $E_1 = r$, $E_2 = re^{-i\phi}$, $E_3 = re^{-i2\phi} \ldots\ldots\ldots E_n = re^{-i(n-1)\phi}$, respectively. The total reflected beam can be concluded as:

$$E = r[1 + e^{-i\phi} + e^{-i2\phi} + \ldots + e^{-i(N-1)\phi}] \tag{3}$$

The total reflected light intensity is then determined to be:

$$I = EE^* = r^2 \frac{\sin^2(Nn_2 k_0 L)}{\sin^2(n_2 k_0 L)} \tag{4}$$

We calculated the reflection spectra of a cascaded Fabry–Perot cavity using Matlab software. During the calculation, we set the amplitude reflection coefficient $r$ to 0.005. Figure 1c,d show the corresponding reflection spectra for 5 and 20 mirrors, respectively. It can be observed that the reflection spectrum exhibits a series of periodically arranged

resonant peaks, and with an increase in the number of reflection mirrors, the intensity of individual resonant peaks increases while the full width at half maximum (FWHM) further narrows, which is a typical phenomenon of multi-beam interference. Figure 1b shows the cross-sectional diagram of the SCF used in this study. Mark each core with the numbers 1–7. When the fiber sensor is bent within a specific fiber orientation angle $\theta$, which is defined as the angle between the bending orientation and the direction of 0°, the reflection spectra of each side core will experience drift due to compression or expansion of the core; $\theta_i$ denotes the angle between 0° and $i$th core.

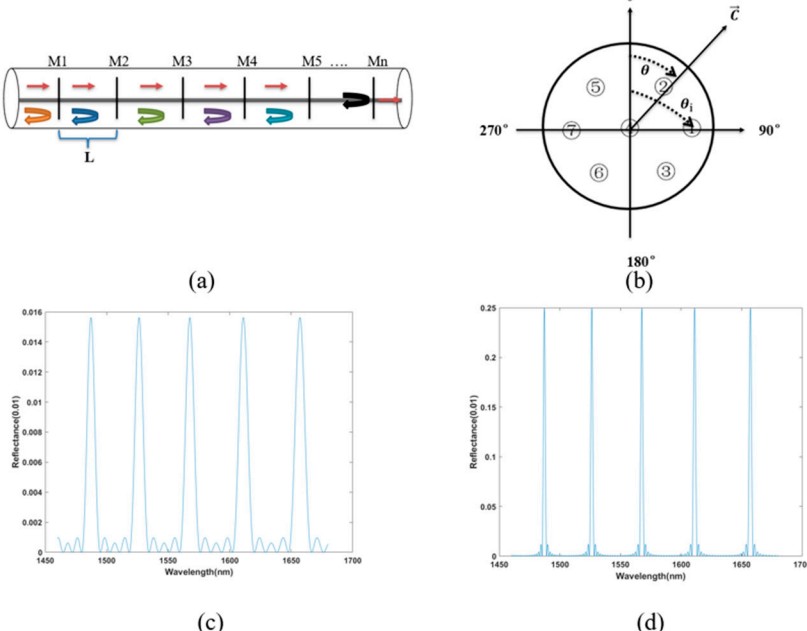

**Figure 1.** (**a**) Light path of one of the SCF cores after inscribing the reflector; (**b**) The cross-sectional diagram of the SCF used in this study; (**c**) Simulated output spectrum of the device for N = 5; (**d**) N = 20.

Figure 2 shows the cross-sectional image of the seven-core optical fiber captured by the CCD. As can be seen, the six-side cores are distributed in a hexagonal shape with a core diameter of 6.1 μm, and the spacing between all cores is 35 μm. Figure 2b shows a schematic of the fiber processing, where a femtosecond laser is focused at the center of the cores in order to create the micro-mirrors. The femtosecond laser used in this experiment was an ytterbium-doped, solid-state laser system (Newport Corp. (Irvine, CA, USA), Spirit One 1040-8-SHG) with a central wavelength of 520 nm. A combination of half-wave plate (HWP) and the Glan-Taylor prism (GTP) was used for controlling the pulse energy. The laser was focused by a 40× objective lens (NA = 0.75) with a repetition rate of 200 kHz. During the inscription, the SCF was fixed on a precision, three-dimensional, electrically controlled displacement platform and immersed in refractive index matching oil to eliminate spherical aberration. In order to achieve a higher reflectivity of each reflector, we employed a line-by-line inscription method to increase the overlap between the fiber core mode and the reflector. The scanning speed during the inscription was set to be 100 μm/s. Specifically, a horizontal line was inscribed along the x-direction to form each reflector. The linewidth of each reflector was 18 μm. This width ensures a high coupling efficiency without affecting the adjacent cores. The line-by-line inscribed mirror was tested with a high back-coupling efficiency of the mode energy, while a fast preparation speed could be guaranteed. The total fabrication time for each cascaded FPs is controlled to be below 10 s. Core 1 and Core 6 were selected for the preparation of cascaded F–Ps. The spatial period of the reflectors was 15 μm, and a total of 100 reflectors were deployed.

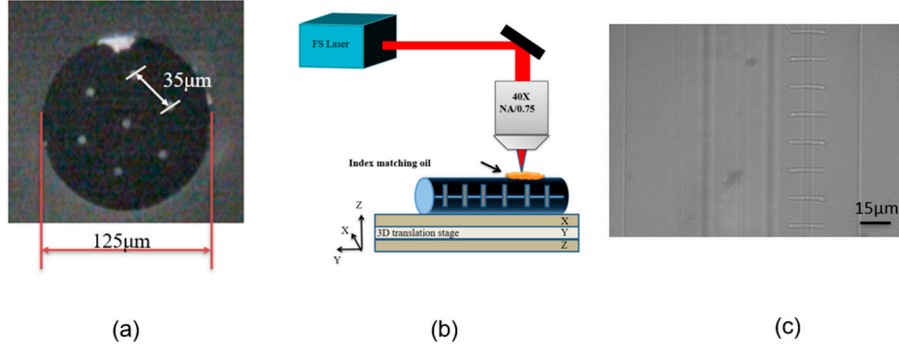

(a)　　　　　　　　　　　　　　(b)　　　　　　　　　　　　　　(c)

**Figure 2.** (**a**) Cross-sectional view of the SCF taken under the CCD; (**b**) schematic diagram of the fabrication apparatus; (**c**) SCF with inscribed reflector under the CCD.

## 3. Sensing Performance Testing and Analysis

We investigated the bending response of the sensor using the setup in Figure 3, where the beam of a broad-spectrum light source (SLD), with a wavelength range between 1250 nm and 1650 nm, is introduced into a cascaded F–P setup through a 3-dB coupler and a Fan-in/Fan-out device, and the output reflection spectrum is monitored in real time by using an optical spectrum analyzer (OSA) with a resolution of 0.05 nm. The curvature of the seven-core fiber can be changed by controlling the distance of the stepper motor displacement stage. The top right corner is a diagram of the SCF bend in a 90° direction.

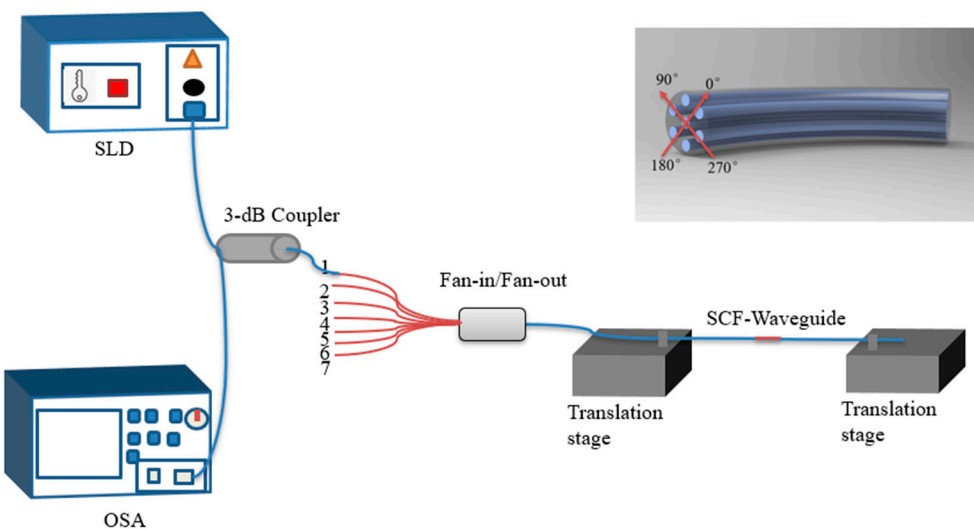

**Figure 3.** Experimental setup for bending response sensing.

Figure 4a,b show the reflection spectra of the cascaded Fabry–Perot cavity prepared in Cores 1 and 6, respectively, without bending. It can be observed that, with a selected spatial period of 15 μm, the free spectral range (FSR) of the reflection signal is approximately 27.75 nm. Figure 4c,d are enlarged views of the reflection peaks indicated by the blue arrows in Figure 4a,b, respectively. They show the maximum extinction ratios of Core 1 and Core 6 in the wavelength range of 1400 nm to 1450 nm, reaching 13 dB and 15 dB, respectively, and the FWHM of the peak around 1450 nm are 0.51 nm and 0.58 nm, respectively. This slight difference may come from errors in the preparation process, such as errors in the focusing depth. The noise (black background) can be attributed to the following reasons: additional Fabry–Perot cavity effects may exist, for example, in the F–Ps formed by the facets such as fiber-tip, the interface between Fan-in/Fan-out, seven-core fiber, and also the inscribed mirrors. A similar noise phenomenon could be found in previous reports, such as reference [26,27]. To verity the reproducibility of the device, two other samples with the same inscription parameters were fabricated and tested. The peak positions were

with no observable difference, with only a slight difference in the level of extinction ratio, showing good repeatability and reliability of this design.

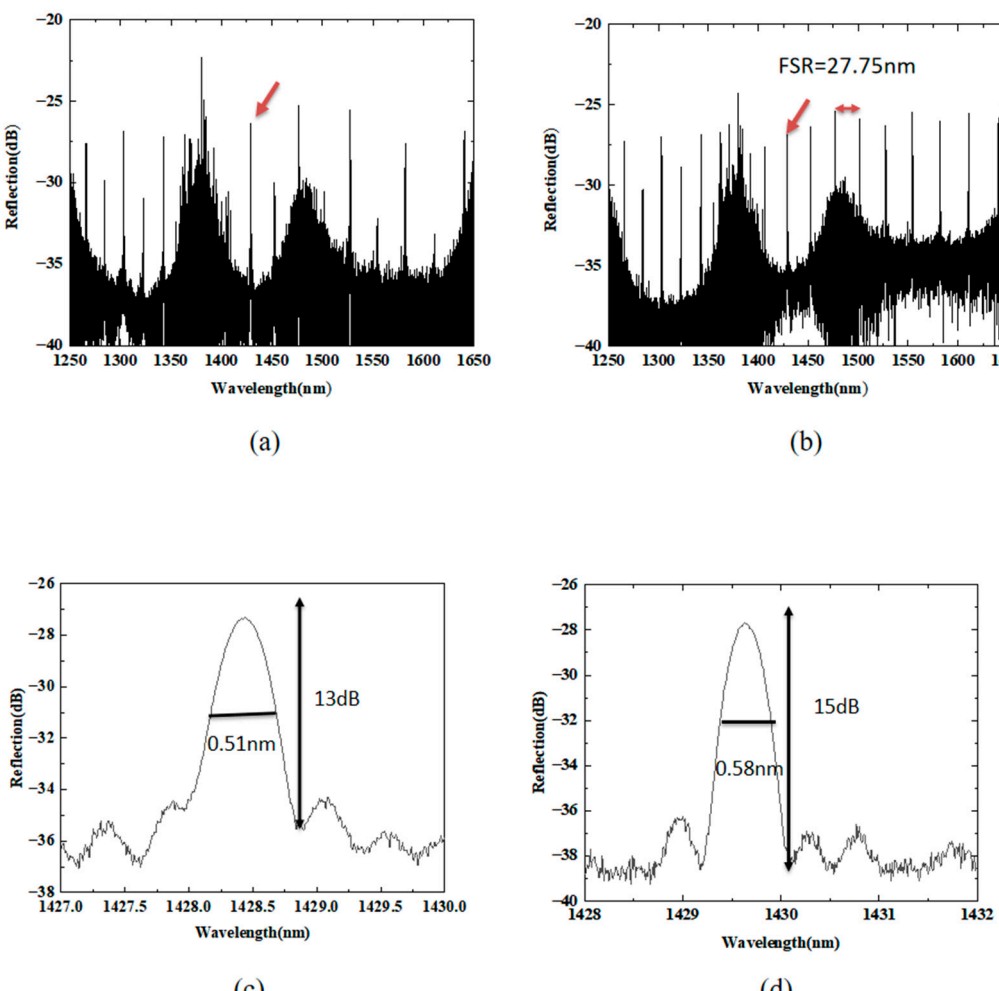

**Figure 4.** Reflection spectrum of reflectors inscribed in (**a**) Core 1; (**b**) Core 6; and (**c,d**) are enlarged views of the reflection peaks indicated by the red arrows in (**a,b**), respectively.

Figure 5a illustrates the experimental setup used for the bending measurement. The two sides of the fiber are fixed on a pair of electrically controlled displacement stages, and the curvature can be controlled by changing the distance between the stages. The control of the bending direction is achieved by pre-rotating the fiber to obtain the desired direction. The curvature $C$ of the bend can be expressed as [28]:

$$C = \frac{1}{R} = \frac{2h}{h^2 + L^2} \tag{5}$$

in which $L$ is half of the distance between the stages, and $h$ is the distance from the horizontal axis. During the test, both sides of the seven-core fiber sample were fixed on the stepper motors with a displacement resolution of 4 nm. The value of $L$ was then obtained by precise control of the displacement of stepper motors. The fiber was attached to a vertically placed acrylic sheet by electrostatic force to keep the fiber bent downwards. The distance h was then measured by using a vernier caliper. A spectrometer was used to record the wavelength shift of the fiber at different curvature and bending orientations. We presented the change of the reflection spectrum with the curvature, as shown in Figure 5b.

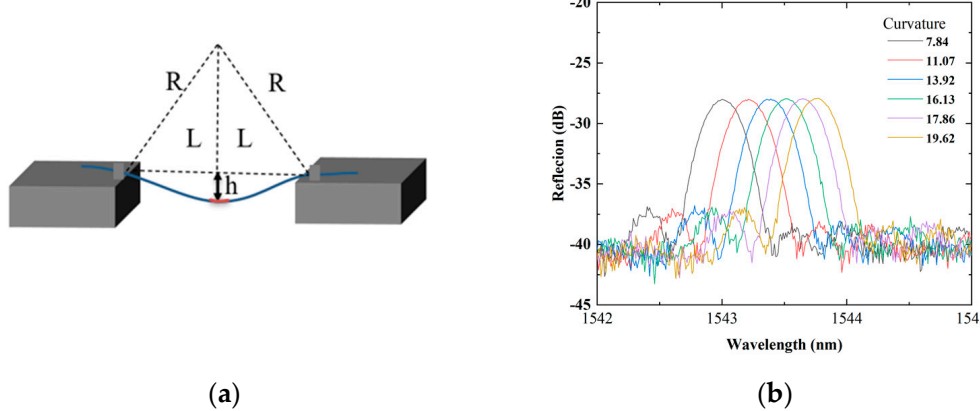

**Figure 5.** (**a**) Bending curvature of an optical fiber; (**b**) the change of the reflection spectrum with the curvature.

Figure 6a,b show the spectral response of the cascaded FPs on Cores 1 and 6 during the test; the resonant peak near 1550 nm was taken as the reference peak. When the fiber is subjected to bending in eight different directions, we can clearly see that for Core 1, the spectrum shifts to the longer wavelength when the SCF bends with the azimuthal angle $\theta$ at 0°, 45°, 90° and 135°, and to the short wavelength direction when the seven-core fiber is subjected to increasing bends at 180°, 225°, 270° and 315°. The bending sensitivity under eight bending conditions are 52.65, 100.19, 113.78, 33.11, −67.08, −90.86, −123.12, and −26.32 pm/m$^{-1}$, respectively. A highest bending sensitivity is achieved at angle $\theta$ of 90°. For Core 6, the spectrum shifts to a longer wavelength when the seven-core fiber is bent at 135°, 180°, 225° and 270°, and to shorter wavelength when the seven-core fiber is bent at 0°, 45°, 90° and 315° when increasing. The bending sensitivities under eight bending conditions are −115.48, −110.38, −36.51, 56.89, 112.08, 118.87, 42.46, and −99.35 pm/m$^{-1}$, respectively. A highest bending sensitivity is achieved at angle $\theta$ of 225°. Figure 6c,d present the dependence of the reflection peak shift of Core 1 and Core 6 on the azimuthal angle $\theta$. It can be observed that the shift grows with the increase of bending curvature and follows a sinusoidal curve with respect to the azimuthal angle $\theta$. The spectral response is most significant when the deformation caused by the azimuthal angle is maximum.

In order to obtain a better understanding of the sensing performance of the SCF, in Figure 7, we present the dependence of the bending sensitivity of Core 1 and Core 6 on the azimuthal angle $\theta$. It can be observed that the sensitivity of each core follows a sinusoidal function with respect to the azimuthal angle, and due to the different orientations of Cores 1 and 6 relative to the fiber axis, the angles corresponding to their maximum sensitivity also differ. Based on the spectral response differences, we can simultaneously determine the curvature and direction of bending by analyzing the reflection spectra of the two cores.

The temperature response of the proposed cascaded F–Ps are also studied, and the results are shown in Figure 8. During the test, the resonant peak near 1550 nm was taken as the reference peak. The SCF was placed in a thermostatic furnace, and the temperature range was controlled between 40 °C and 70 °C with a step size of 6 °C. In each step of the test, spectrum was captured after the temperature was kept constant for 20 min to minimize the influence of temperature fluctuations. It can be seen that for both cores, the spectral shift shows a linear relationship with temperature, and the temperature sensitivities of Core 1 and Core 6 are at a similar level (8.33 pm/°C and 8.13 pm/°C, respectively). The sensitivity obtained is similar to the results reported in the previous literature, indicating that the sensor can also be used for temperature detection. Due to the different spectral response characteristics of the two cores, both bending and temperature sensing can be achieved simultaneously.

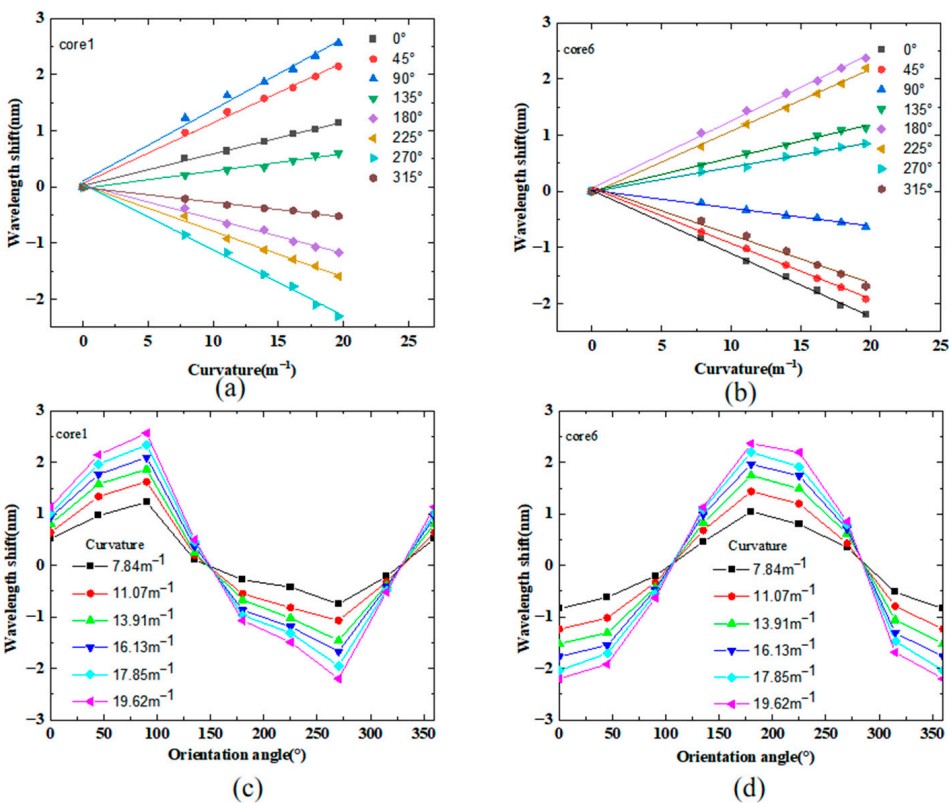

**Figure 6.** Offsets of tilt wavelengths relative to curvature measured at different orientations: (**a**) core1 and (**b**) core6 (**c**) offset of tilt wavelengths in Core 1 and (**d**) Core 6, plotted as a function of azimuth angle.

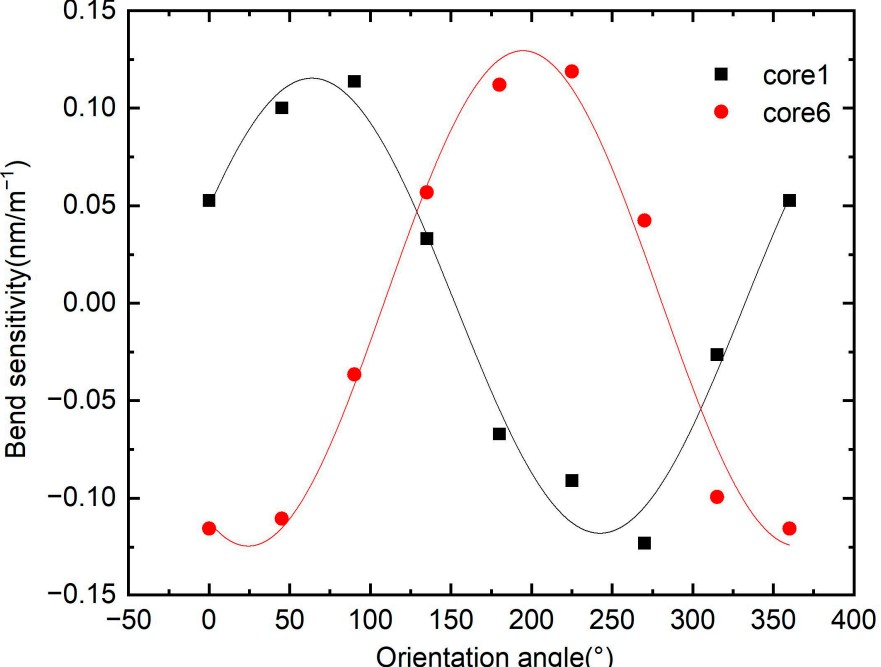

**Figure 7.** Curvature sensitivity in relation to azimuth angle.

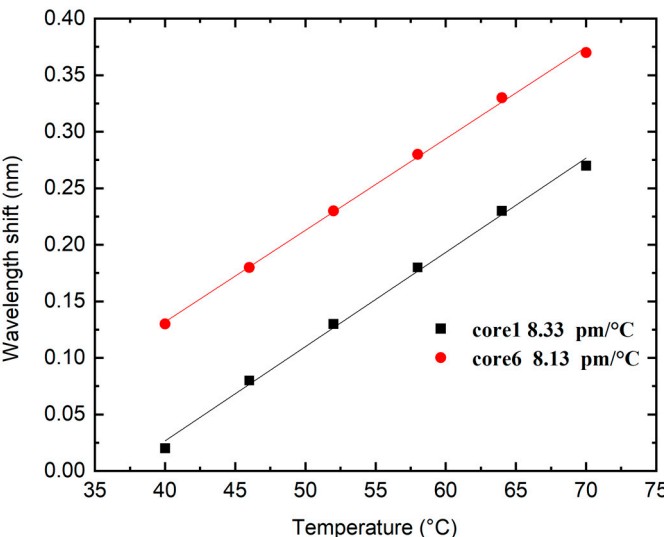

**Figure 8.** Read the temperature response of the proposed cascaded F–Ps.

## 4. Curvature Reconstruction

The correlation between the orientation angle and curvature sensitivity of each outer core F–P, as shown in Figure 7, is fitted with a sinusoidal function as presented in below:

$$S_{\mathrm{i}} = y_i + A_i \sin(\theta - \theta_i) \tag{6}$$

in which $S_i$ denotes the curvature sensitivity in the $\theta$ direction, and $y_1 = -0.001$, $A_1 = 0.117$, $y_6 = 0.001$, and $A_6 = 0.129$ are constants which can be obtained from Figure 7.

For any specific two-dimensional bending condition, the curvature sensitivity is written as:

$$S_{\mathrm{i}} = \frac{\Delta \lambda_i}{C} \tag{7}$$

in which $\Delta \lambda_i$ stands for the measured peak wavelength shift of core *i*.

Combining Equations (6) and (7), the relationships between curvature *C*, bending orientation $\theta$ and the wavelength shift can be derived as:

$$\begin{cases} C = \dfrac{\Delta \lambda_1}{y_1 + A_1 \sin(\theta - \theta_1)} \\ C = \dfrac{\Delta \lambda_6}{y_6 + A_6 \sin(\theta - \theta_6)} \end{cases} \tag{8}$$

in which $\Delta \lambda_1$ and $\Delta \lambda_6$ are the wavelength shift measured from Cores 1 and 6, respectively. By applying Equation (8), the curvature *C* and the orientation $\theta$ can be reconstructed once the wavelength shift is known. For a specific set of conditions ($\theta = 45°$, $C = 11.07$ m$^{-1}$) $\Delta \lambda_1$ and $\Delta \lambda_6$ are 1.34 nm and $-1.02$ nm, respectively, and the curvature magnitude C and azimuth angle $\theta$ can be reconstructed by Equations (6)–(8). The reconstructed results using Fiber Core 1 and Fiber Core 6 are $C = 10.55$ m$^{-1}$, $\theta = 44.89°$. The relative error of curvature is 4.6%, and the average relative deviation of the azimuth angle $\theta$ is about 0.2%.

## 5. Conclusions

We propose and manufacture a two-dimensional vector bending sensor which uses a reflector embedded in a seven-core fiber and a femtosecond laser to write a cascade fiber grating in the fiber core line by line. The line-by-line inscription strategy applied shows fast processing speed with a total fabrication time of each cascaded FPs below 10 s, while a high back-coupling efficiency of the core energy can be realized. The bending response of two reflectors on an asymmetric outer core is studied in a 360° direction with a step size of 45°. The maximum sensitivity is $-123.12$ pm/m$^{-1}$. In bending tests, the directional behavior of the light response can be observed due to the geometry of the fiber. The direction and

magnitude of curvature can be reconstructed by using the wavelength drift and the regular hexagonal geometric relationship between the cores. Under certain conditions ($\theta = 45°$, $C = 11.07 \text{ m}^{-1}$), the relative error of curvature is 4.6% and the mean relative deviation of the azimuth Angle $\theta$ is about 0.2%. In addition, we tested the temperature of lettering fiber, and the change of wavelength drift is not as significant as the change of bending when the temperature rises. The fabricated cascaded fiber Bragg grating vector bending sensor has the advantages of compact structure, fast fabrication and also strong reproducibility, which is expected to realize real-time monitoring of the bending curvature and bending direction of intelligent engineering structures.

**Author Contributions:** Conceptualization, Y.Y.; methodology, L.Z.; software, Y.Y.; investigation, Y.Z.; writing original draft preparation, Y.Z.; writing review and editing, H.M. supervision, Y.Y., M.W., Z.N. and C.B.; funding acquisition, Y.Y., L.Z., Z.N. and C.B. All authors have read and agreed to the published version of the manuscript.

**Funding:** This research was funded by Natural Science Foundation of China, grant number 11774071; by the Natural Science Foundation of Shandong Province, grant number ZR2020QF86, ZR2022MF253; and by Liaocheng University, grant number 318051411, 318052199.

**Institutional Review Board Statement:** Not applicable.

**Informed Consent Statement:** Not applicable.

**Data Availability Statement:** Data available on request.

**Acknowledgments:** Partial financial supports from the Natural Science Foundation of Shandong Province and Liaocheng University, Shandong, China, are highly appreciated.

**Conflicts of Interest:** The authors declare no conflict of interest.

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
