# Peer review of "Femtosecond Laser Line-by-Line Inscribed Seven Core Fiber Cascaded Fabry–Perot Cavity and Its Vectorial Bending Sensing Application"

_photonics, doi:10.3390/photonics10060605_

Round 1

Reviewer 1 Report

This paper presents a two-dimensional vector bending sensor based on ultrafast laser inscribed multiple sets of reflectors (Fabry-Perot structure) in a multicore fiber (seven-core). Both bending direction and curvature magnitude can be reconstructed from the spectral displacement of any two non-diagonal outer core fiber gratings. Besides of vector bending sensing in any direction, real-time measurement of temperature can be realized as well. Overall, the manuscript is prepared with lots of useful details, which might be very helpful for the researchers working in this area. The manuscript could be considered for publication after addressing the following questions.

1. Please add more details to the Figures, i.e., extinction ratios and FSR in Figures 4, FWHM as a zoomed-in inset in Figures 4, as well as sensitivities in Figure 8.

2. Please proofread and correct typos and formats in figure captions, i.e., Figure 8.

3. Please correct typos/errors in lines 145 & 150 on Page 5 and lines 202 & 211 on Page 8.

4. Could the authors share more information about the seven-core fiber Fan in/Fan out device? Is there any mode-field diameter mismatch that may affect the loss performance of extinction ratio of the reflection spectra?

5. Could the authors explain how to calculate curvature precisely? Will the measurement uncertainties of h and L affect the bend sensitivities?

Reviewer 2 Report

The authors report on the fabrication of a seven core fiber in which micro-mirrors have been fabricated by femtosecond laser writing. These mirrors act as cascaded Fabry-Perot cavity, and the device is proposed as vectorial bending sensor.

The work is interesting for researchers working in the field fiber-based stress sensor. However, some points must be addressed before publication.

1.      As far as the fabrication is concerned how many mirrors have been fabricated in the device under test? And how long is the writing region?

2.      Do the mirror cover any of the seven core or just part of the cladding?

3.      What is the origin of noise (black background of the curves) in the reflection spectra shown in figure 4?

4.      What is the analytical procedure to retrieve the curvature and the orientation from the measured spectra?

5.      What is the effect of a stress (compression or elongation) along the fiber direction. i.e. without curvature?

6.      Can the author comment on the reproducibility of such a device? How much the calibration curves shown in figure 6 depend on the position (respect the cores) and dimension of the mirrors (fixed their spacing)?

Reviewer 4 Report

This paper proposes a directional bending sensor based in cascade Fabry-Perot interferometers fabricated in the cores of a seven-core fiber. However, The use of seven core fiber with cascade FPIs as curvature sensors has been proposed with better sensitivity ( ~48 nm/m-1 ) (Aoao Yang, Weijia Bao, Fengyi Chen, Xingyong Li, Ruohui Wang, Yiping Wang, and Xueguang Qiao, "Two-dimensional displacement (bending) sensor based on cascaded Fabry–Perot interferometers fabricated in a seven-core fiber," Opt. Express 31, 7753-7763 (2023)) so there is no significant contribution in this paper. At least the authors should comment on this work, and clearly state the novelty of this paper. Furthermore, the manuscript lacks sufficient technical quality. The spectral response of the cascaded FPIs presented are very noise and unclear, and are only presented for the case without bending, so it is not clear how they change with the curvature.

Round 2

Reviewer 1 Report

The revised version is in good shape and could be considered to publish in Photonics. 

Reviewer 2 Report

The authors have addressed all the raised points of my review. The paper can be accepted in the present form

Reviewer 4 Report

I'm satisfied with the authors corrections. So, I recommend this paper for publication.